# Exploring the Antibacterial and Antiparasitic Activity of Phenylaminonaphthoquinones—Green Synthesis, Biological Evaluation and Computational Study

**DOI:** 10.3390/ijms251910670

**Published:** 2024-10-03

**Authors:** Sussan Lopez-Mercado, Cinthya Enríquez, Jaime A. Valderrama, Ricardo Pino-Rios, Liliana Ruiz-Vásquez, Lastenia Ruiz Mesia, Gabriel Vargas-Arana, Pedro Buc Calderon, Julio Benites

**Affiliations:** 1Magister en Ciencias Químicas y Farmacéuticas, Facultad de Ciencias de la Salud, Universidad Arturo Prat, Casilla 121, Iquique 1100000, Chile; sulopez_@estudiantesunap.cl (S.L.-M.); rpinorios@unap.cl (R.P.-R.); 2Doctorado en Química Medicinal, Facultad de Ciencias de la Salud, Universidad Arturo Prat, Casilla 121, Iquique 1100000, Chile; cenriquez@estudiantesunap.cl (C.E.); pedro.buccalderon@uclouvain.be (P.B.C.); 3Laboratorio de Química Medicinal, Química y Farmacia, Facultad de Ciencias de la Salud, Universidad Arturo Prat, Casilla 121, Iquique 1100000, Chile; jaimeadolfov@gmail.com; 4Centro de Investigación de Recursos Naturales, Universidad Nacional de la Amazonía Peruana (UNAP), AA. HH. “Nuevo San Lorenzo”, Pasaje Paujiles S/N, San Juan Bautista, Iquitos16002, Peru; liliana.ruiz@unapiquitos.edu.pe (L.R.-V.); lastenia.ruiz@unapiquitos.edu.pe (L.R.M.); 5Facultad de Farmacia y Bioquímica, Universidad Nacional de la Amazonía Peruana, Nina Rumi, San Juan Bautista, Iquitos 16000, Peru; 6Laboratorio de Química de Productos Naturales, Instituto de Investigaciones de la Amazonía Peruana (IIAP), Av. Abelardo Quiñones km 2.5, Iquitos 16001, Peru; 7Facultad de Industrias Alimentarias, Universidad Nacional de la Amazonía Peruana, Zungarocha S/N, San Juan Bautista, Iquitos 16002, Peru; 8Research Group in Metabolism and Nutrition, Louvain Drug Research Institute, Université Catholique de Louvain, 73 Avenue E. Mounier, 1200 Brussels, Belgium

**Keywords:** phenylaminonaphthoquinones, amination reaction, solvent-free reaction, silica gel, antimicrobial, antiplasmodial

## Abstract

Organic compounds with antibacterial and antiparasitic properties are gaining significance for biomedical applications. This study focuses on the solvent-free synthesis (green synthesis) of 1,4-naphthoquinone or 2,3-dichloro-1,4-naphthoquinone with different phenylamines using silica gel as an acid solid support. The study also includes in silico PASS predictions and the discovery of antibacterial and antiparasitic properties of phenylaminonaphthoquinone derivatives **1**–**12**, which can be further applied in drug discovery and development. These activities were discussed in terms of molecular descriptors such as hydrophobicity, molar refractivity, and half-wave potentials. The in vitro antimicrobial potential of the synthesized compounds **1**–**12** was evaluated against a panel of six bacterial strains (three Gram-positive: *Staphylococcus aureus*, *Proteus mirabilis*, and *Enterococcus faecalis*; and three Gram-negative bacteria: *Escherichia coli*, *Salmonella typhimurium*, and *Klebsiella pneumoniae*). Six compounds (**1**, **3**, **5**, **7**, **10,** and **11**) showed better activity toward *S. aureus* with MIC values between 3.2 and 5.7 μg/mL compared to cefazolin (MIC = 4.2 μg/mL) and cefotaxime (MIC = 8.9 μg/mL), two cephalosporin antibiotics. Regarding in vitro antiplasmodial activity, compounds **1** and **3** were the most active against the *Plasmodium falciparum* strain 3D7 (chloroquine-sensitive), displaying IC_50_ values of 0.16 and 0.0049 μg/mL, respectively, compared to chloroquine (0.33 μg/mL). In strain FCR-3 (chloroquine-resistant), most of the compounds showed good activity, with compounds **3** (0.12 μg/mL) and **11** (0.55 μg/mL) being particularly noteworthy. Additionally, docking studies were used to better rationalize the action and prediction of the binding modes of these compounds. Finally, absorption, distribution, metabolism, excretion, and toxicity (ADMET) predictions were performed.

## 1. Introduction

Infectious diseases caused by pathogenic microorganisms (e.g., bacteria, viruses, parasites, and fungi) are responsible for mortality worldwide and constitute a global health problem [1]. Since 1990, it has been estimated that over 10 million people have died due to infections [2]. The World Health Organization (WHO) predicts that infectious diseases will cause 10 million deaths annually by 2050 [3]. Although the incidence of infectious diseases has decreased globally, it continues to pose a serious public health problem that has resulted in significant social and economic hardships [4].

Addressing this global challenge requires the development of effective therapeutic agents, and 1,4-naphthoquinones containing an amino group have emerged as valuable compounds in medicinal chemistry. These compounds have demonstrated a wide range of applications, including antibacterials, antimalarials, antituberculars, larvicides and molluscicides, herbicides, and fungicides [5,6,7,8,9]. Figure 1 shows some substituted 1,4-naphthoquinones that have been reported as potential antibacterial agents [10,11,12,13,14,15,16,17]. The mechanism of these activities lies in the generation of reactive oxygen species (ROS) leading to apoptotic cell death [18,19,20]. These compounds have not only served as models for improving clinically available antibacterial agents but have also provided insights into the mechanisms of action that are not yet fully understood [21].

Parasitic diseases such as malaria, a disease caused by Plasmodium parasites, continue to affect millions of people annually, with *Plasmodium falciparum* and *Plasmodium vivax* posing significant public health threats [22,23,24]. Figure 2 shows some aminonaphthoquinones displaying antimalarial activity including their IC_50_ values and percentage of parasitemia inhibition [25,26,27,28,29,30,31,32]. Several mechanisms have been proposed to explain such activity, among them competitive inhibition of the cytochrome bc1 complex, generation of ROS, enzymatic inhibition (e.g., glutathione reductase, dihydroorotate dehydrogenase, and glycerol glyceraldehyde-3-phosphate dehydrogenase), alkylation of biomolecules, and depletion of glutathione [33].

Given the critical role of these compounds in combating infectious diseases, it is essential to explore sustainable synthesis methods for their production. The green chemistry initiative has introduced metrics such as atom economy, E factor, and process mass intensity to promote greener and more sustainable chemical processes in both industrial and academic sectors [34]. In particular, solvent selection is crucial for designing efficient synthetic strategies, as solvents contribute significantly to material use, energy consumption, and environmental impact in pharmaceutical production [35]. Solvent-free synthesis methods offer a promising alternative by eliminating volatile organic compounds, resulting in safer, more environmentally friendly processes with simpler execution and higher yields [36].

The design of solvent-free reactions has been the subject of considerable interest in the field of green synthesis and is a key focus of this work. Herein, we report the efficient conjugate addition of phenylamines to naphthoquinones without any solvent, using silica gel as an acid solid support. The study also includes in silico PASS predictions. The antimicrobial and antiplasmodial activities of the resulting compounds were investigated, and they were further correlated with some molecular descriptors. Molecular docking analysis against the clumping factor A from *Staphylococcus aureus*, the atovaquone-inhibited cytochrome bc1 complex, and dihydroorotate dehydrogenase were also conducted as well. Additionally, the pharmacokinetic properties related to absorption, distribution, metabolism, excretion, and toxicity (ADMET) were calculated using the pkCSM online tool.

## 2. Results

### 2.1. Solvent-Free Synthesis

The process of optimizing the reaction conditions for the amination reaction of 1,4-naphthoquinone with phenylamines proceed through an initial irreversible aza-Michael addition reaction. The green synthetic method used was simple, utilizing an ancient tool for mechanochemistry: an agate mortar and pestle, known as “grindstone chemistry”. This was achieved through a model reaction carried out under solvent-free conditions in the absence of silica gel at room temperature. The adduct yield after 15 h was 65%. Additionally, a similar reaction conducted in the presence of silica gel at room temperature, where 1,4-naphthoquinone (1 mmol) and p-hydroxyaniline (1 mmol) were supported on silica gel (0.2 g), was completed after 30 min under solvent-free conditions, resulting in a 89% isolated yield of the phenylaminonaphthoquinone **5** (Table 1). Spent silica gel was easily recycled after simple filtration, washed with acetone, and dried at 100 °C. To demonstrate the substrate scope of this method, similar reaction conditions were applied to the reactions of 1,4-naphthoquinone and 2,3-dichloro-1,4-naphthoquinone compounds with different phenylamines the experimental details, isolation and structural characterization of the resulting phenylaminonaphthoquinone in the assays have been reported by our research group. Table 1 compares the yields and reaction times under solvent-free conditions versus those using ethanol as solvent at room temperature. The results showed that similar yields are obtained in both types of reactions, but a shorter reaction time is required for reactions without solvent than those performed with ethanol.

### 2.2. The Prediction of Biological Activity Based on Spectra

The prediction of biological activity based on spectral analysis of synthesized compounds in this study was accomplished using the freely available PASS and PharmaExpert tools, accessible at: https://www.way2drug.com/PassOnline/ (accessed on 26 March 2024). The activity spectrum for a molecule provides a list of biological activities along with the probabilities of the compound being active (Pa) and inactive (Pi), with values ranging from 0 to 1. Compound **3** was selected as an example, and its PASS prediction results are shown in Table 2. For antibacterial activity, compound **3** showed high Pa as a membrane permeability inhibitor (Pa = 0.700) and p-aminobenzoic acid antagonist (Pa = 0.536), while as an anti-infective, antibiotic anthracycline-like, and antibacterial, it demonstrated Pa values of 0.312, 0.192, and 0.162, respectively. The antiprotozoal activity had a Pa of 0.174. The Pi values were low for all activities except for antibacterial activity.

### 2.3. Antimicrobial Activity

We evaluated the antibacterial activity of phenylaminonaphthoquinones **1**–**12**, (Table 3) against three Gram-positive (*Staphylococcus aureus*, *Proteus mirabilis*, and *Enterococcus faecalis*) and three Gram-negative (*Escherichia coli*, *Salmonella typhimurium*, and *Klebsiella pneumoniae*) human pathogens, which exhibit morphological and physiological differences.

As shown in Table 2, phenylaminonaphthoquinones **1**–**12** displayed antibacterial activity against Gram-positive bacteria, particularly *Staphylococcus aureus*. Compounds **1**, **3**, **5**, **7**, and **11** showed good activity against *S. aureus*, with compounds **1**, **3,** and **5** highlighting MIC values between 3.2 and 3.9 μg/mL, which are lower compared to the antibacterial drugs cefazolin (MIC = 4.2 μg/mL) and cefotaxime (MIC = 8.9 μg/mL). Interestingly, compound **9** loses activity against *S. aureus*. Among the tested series, halogenated derivatives (2-chloro-3-phenylaminonaphthoquinones **2**, **4**, **6**, **8**, and **12**) showed the least increase in activity against *S. aureus,* with the exception of compound **10.**

In contrast, phenylaminonaphthoquinones **1**–**12** did not exhibit antibacterial activity against three Gram-negative bacteria (*E. coli*, *S. typhimurium*, and *K. pneumoniae*).

### 2.4. Antiplasmodial Activity

Table 4 shows the growth inhibition percentages and IC_50_ values of chloroquine (CQ) and phenylaminonaphthoquinone derivatives **1**–**12** against *Plasmodium falciparum* 3D7 (chloroquine-sensitive) and FCR-3 (chloroquine-resistant). Compounds **1**, **3**, and **12** showed antiplasmodial activity against *Plasmodium falciparum* 3D7, and compounds **1**, **3**–**7**, and **11** exhibited antiplasmodial activity against *Plasmodium falciparum* FCR-3.

The compounds showing a percentage of growth inhibition equal to or greater than 50% were subjected to serial dilutions to calculate their IC_50_ values. These included compounds **1**, **3**, and **12** against *Plasmodium falciparum* 3D7, and compounds **1**, **3–7** and **11** against *Plasmodium falciparum* FCR-3 as shown in Table 3. Chloroquine (CQ) was used as a standard antiplasmodial drug for comparison.

Regarding IC_50_ values, the results showed that compound **3** was the most active, followed by compounds **1** and **12**, against the *Plasmodium falciparum* strain 3D7, displaying IC_50_ values of 0.0049, 0.16, and 0.80 μg/mL, respectively, compared to CQ, which had an IC_50_ value of 0.06 μg/mL. In the FCR-3 strain, most of the compounds exhibited good activity, with compounds **3** (0.12 μg/mL) and **11** (0.55 μg/mL) being particularly noteworthy.

### 2.5. Physicochemical Descriptors

Three standard molecular descriptors commonly utilized in structure–activity relationships were evaluated: hydrophobicity (LogP), steric effect as molar refractivity (CMR, expressed in cm^3^/mol), and half-wave potential (−E^I^_1/2_, expressed in mV). Table 5 shows the experimental and theoretical LogP values, demonstrating a good correlation between them. The molar refractivity (CMR) of the phenylaminonaphthoquinone compounds was systematically lower than that of those bearing a chlorine substituent on the naphthoquinone ring. The half-wave potential (−E^I^_1/2_) ranges between −856 and −727 mV for phenylaminonaphthoquinones compounds **1**, **3**, **5**, **7**, **9**, and **11.** In contrast, compounds **2**, **4**, **6**, **8**, **10**, and **12**, which contain chlorine substituents at the 2-position, had more positive values ranging between −690 and −530 mV.

### 2.6. Molecular Docking

Table 6 shows the free binding energy values for the ligand–protein complexes with the phenylaminonaphthoquinone compounds. Three types of proteins were evaluated for this in silico study: clumping factor A from *Staphylococcus aureus* (1N67) for antimicrobial activity, and atovaquone-inhibited cytochrome bc1 complex (4PD4) and dihydroorotate dehydrogenase (5FI8) for antiplasmodial activity. The results of these docking experiments showed that compounds **1**–**12** displayed higher interaction with the clumping factor A from *Staphylococcus aureus,* particularly the halogenated derivatives (2-chloro-3-phenylaminonaphthoquinones). For the antiplasmodial activity, it was observed that there was a better interaction with the target protein dihydroorotate dehydrogenase of *Plasmodium falciparum* (PDB ID: 5FI8) compared to atovaquone-inhibited cytochrome bc1 complex (PDB ID: 4PD4). Moreover, compounds **1**–**12** exhibited a better interaction than chloroquine in all cases.

Figure 3(left side) illustrates the optimal location for the interaction between compound **3** and clumping factor A (ΔE_Bind_ = −9.4 kcal·mol^−1^), an index of antibacterial activity. The right side shows a two-dimensional representation of the bonding interactions between the residues and the ligand. Note the formation of hydrogen bonds resulting from a significant number of hydrogen bond acceptor oxygen, Van der Waals interactions, and π (aromatic)–σ interactions. These interactions are robust enough to compensate for unfavorable donor–donor interactions (as indicated in the legend of Figure 3).

On the other hand, the free binding energy that results from molecular docking calculations of phenylaminonaphthoquinones **1**–**12** with the target protein for atovaquone-inhibited cytochrome bc1 complex (antiplasmodial activity) shows similar values, ranging from −9.2 to −8.4 kcal/mol^−1^. Among the compounds tested, the most active in vitro against *Plasmodium falciparum* 3D7 (chloroquine-sensitive) was compound **3**, which displayed a free binding energy of −9.2 kcal/mol. The majority of its attractive interactions are Van der Waals in nature, involving amino acids such as Leu282, Leu276, Phe278, Met295, Ileu147, Phe129, and Gly143, along with alkyl interactions with Tyr279 and Pro271. The interaction of compound 3 with the binding site is also hydrophobic, forming π–δ bonds with three amino acid residues (Ile269, Val146, Leu275), as well as π–π stacking with the amino acid Trp142, as illustrated in the docking report (Figure 4).

Figure 5 shows the dihydroorotate dehydrogenase (DHODH). The molecular docking calculations of phenylaminonaphthoquinones **1**–**12** with DHODH yielded free binding energy values, ranging from −10.2 to −9.1 kcal.mol^−1^. This higher affinity is due to the oxygen atoms in the naphthoquinone ring, which can act as hydrogen bond acceptors and can easily interact with the surrounding amino acid donors. Additionally, the aromatic substituents contribute to the interaction of the complex through Van der Waals interactions. For example, in Figure 5, the best position found for compound **3** with dihydroorotate dehydrogenase (ΔE_Bind_ = −9.1 kcal.mol^−1^) as well as the close interactions that this compound possesses. A hydrogen bond is observed between compound **3** and Phe278 as well as other weak interactions related to the substituent aromatic ring.

### 2.7. ADMET Profiles

Pharmacokinetic parameters and toxicity data were obtained using the pkCSM Online Tool and are reported in Table 7.

ADMET analysis revealed that compounds **1**–**12** had a molecular weight of less than 500 g/mol, including chloroquine, which is crucial for penetrability. All compounds displayed Caco-2 permeability values greater than 1.834 and high intestinal absorption (90.1–97.4%), indicating that they would be absorbed in the small intestine. The transdermal efficacy, as demonstrated by the skin permeability of compounds **1**–**12**, ranged from −2.141 to −2.991 cm/h, suggesting that they would penetrate the skin properly. A VDss value higher than −0.15 is acceptable for compounds, and all of these compounds, with the exception of chloroquine, had log BB values less than 0.3, indicating that they would be able to penetrate the brain. None of the compounds, except chloroquine, were CYP2D6 inhibitors, and only compounds **3**, **5**, and **6** did not inhibit CYP3A4, a potential interference with CYP450 biotransformation reactions. The total clearance values indicated that only compound **6** reached a negative value (−0.065 log mL/min/kg), whereas the rest of the compounds had positive values, suggesting rapid excretion. Finally, the acute oral toxicity in rats (LD_50_) ranged from 1.918 to 2.937 mol/kg, corresponding to toxicity values.

## 3. Discussion

The synthetic evidence of our study demonstrates that the amination reaction of 1,4-naphthoquinones or 2,3-dichloronaphthoquinones with phenylamines supported on silica gel, offering a viable alternative to existing methodologies [38,39,40,41,42,43,44,45,46]. The results indi-cate that the optimized solvent-free reaction conditions are not only straightforward but also highly efficient, representing a significant advancement in green chemistry by elimi-nating the need for organic solvents, which are typically harmful to the environment.

Other works have developed methods of C–N and C–S bond formation by Michael addition of α-aminoamide by conjugate addition of amines to bulky α,β-unsaturated am-ides and of thiols to α,β-unsaturated ketones under solvent-free conditions at 50 °C, all reactions promoted by silica gel as a mild Lewis acid. The yields of the adducts are good to excellent, and the workup of the reaction mixture is simple and not time-consuming [47,48].

The synthesis of the adduct through the reaction of 1,4-naphthoquinone with p-hydroxyaniline under solvent-free conditions yielded 65% after 15 h, without the use of any catalyst or solvent, highlighting the potential of this approach in facilitating chemical transformations. The introduction of silica gel, as an acid solid support medium, resulted in a marked improvement in both reaction time and yield. The reaction was completed in just 30 min, with a significantly higher yield of 89%. This suggests that silica gel plays a crucial role in enhancing the reaction kinetics, possibly by providing a larger surface area for the reactants to interact and by stabilizing transition states, which facilitates the formation of the desired product. Moreover, the recyclability of silica gel, demonstrated through simple washing and drying, enhances the environmental and economic viability of this method. Expanding the scope of the reaction to other phenylamines with 1,4-naphthoquinone or 2,3-dichloro-1,4-naphthoquinone showed similar trends, further underscoring the general applicability of the solvent-free method. Consistently producing yields comparable to those obtained using ethanol as a solvent at room temperature, but with shorter reaction times, this approach suggests that solvent-free synthesis is both versatile and efficient. It offers significant time saving and environmentally friendly benefits for the synthesis of various naphthoquinone derivatives.

The prediction of biological activity using the PASS and PharmaExpert tools adds an additional layer of value to the synthesized compounds by providing insights into their potential pharmacological applications [49,50]. The predicted antibacterial and antiplasmodial activities with *Pa* values between 0.161–0.225 and 0.174–0.266, respectively suggest that these compounds have a moderate likelihood of exhibiting these activities. While these predictions are valuable for guiding further experimental work, it is crucial to consider factors such as the compound’s bioavailability, toxicity, interaction with biological targets, specific interests in certain types of activities, desirable novelty, and available facilities for experimental testing, as these aspects can significantly influence the actual biological activity and the overall planning of experiments [51].

Based on the PASS and antibacterial action mechanism of quinones [5,17,52,53], the evaluation of the antimicrobial activities of phenylaminonaphthoquinones **1**–**12** revealed promising results, particularly against *Staphylococcus aureus*, a notorious human pathogen associated with a wide range of infections, and well-known for its resistance to several antibiotics [54].

The data in Table 2 were examined to establish a preliminary structure-activity relationship (SAR) analysis. The study, based on the nature of substituents located around the 2-phenylamino-1,4-naphthoquinone core of the pharmacophore (compound **1**) indicate that the insertion of the methyl group in 2”-position of the arylamino substituent increase the antibacterial activity in two of the three Gram-positive strains and in all three Gram-negative strains. In contrast, insertion of oxygen group(s) in the arylamino substituent of the scaffold, as seen in compounds **5**, **7**, and **9**, did not produce significant changes in biological activity. Thus, insertion in 4′-position of hydroxyl or methoxy groups in the pharmacophore, as in compounds **5** and **7** induce minimal effect on the activity, resulting only in slight variations in specific cases. The insertion of two methoxy groups at 2′,5′-positions in the pharmacophore, generally caused a decrease in antibacterial activity. Finally, the insertion of a methyl group bonded to the nitrogen atom of the scaffold, as in compound **11**, led to a reduction in antibacterial activity in two of the three tested Gram-positive and Gram-negative strains. Interestingly, the series of halogenated derivatives compounds **2**, **4**, **6**, **8**, and **12** display lower antibacterial activity against one Gram-positive strain (*S. aureus*), except for compound **10**. To note that halogenation does not necessarily enhance antibacterial activity, warranting further investigation into the SAR that govern these effects.

In general, compounds **1**, **3**, and **5** showed superior activity against *S. aureus*, with lower MIC values compared to standard antibiotics such as cefazolin and cefotaxime. This suggests that these phenylaminonaphthoquinones could be potential candidates for de-veloping new antibacterial agents against *S. aureus*, addressing the growing concern re-garding antibiotic resistance. Medina et al. [55] demonstrated that aminonaphthoquinones compounds induce an increase in ROS, which are toxic to *S. aureus* cells. Additionally, they suggested that the amino group in these compounds may be ionizable, reducing its ability to permeabilize membranes and inhibit *S. aureus* growth. Other Gram-positive strains such as *P. mirabilis* and *E. faecalis*, were less susceptible to the effects of the tested compounds.

In contrast, the phenylaminonaphthoquinones **1**–**12** displayed no antibacterial activ-ity against the Gram-negative bacteria tested (*E. coli*, *S. typhimurium*, and *K. pneumoniae*). Several authors have suggested that hydrophobicity (LogP) plays a crucial role in the an-tibacterial activity of compounds, with a complex relationship that varies between Gram-positive and Gram-negative bacteria [56,57]. Although increased lipophilicity generally enhances penetration through the lipid-rich outer membrane of Gram-negative bacteria, the optimal LogP value for activity appears to differ between bacterial classes [58]. However, this parameter did not seem to influence the activity of the phenylaminonaphthoquinones. The lack of activity observed against Gram-negative bacteria could be attributed to the structural and physiological differences between Gram-positive and Gram-negative bacteria, particularly the more complex and impermeable outer membrane of Gram-negative bacteria, which often makes them less susceptible to certain antimicrobial agents [58]. Although the precise mechanism of action of phenylaminonaphthoquinones remains incompletely elucidated, it is hypothesized that these compounds induce oxidative stress and disrupt cellular membranes.

The phenylaminonaphthoquinones also demonstrated notable antiplasmodial activity, particularly against *Plasmodium falciparum*, a major causative agent of malaria. This activity aligns with the known potential of quinone-derived compounds in antimalarial drug discovery [59]. Among the 12 synthesized compounds **1**, **3**, and **12** exhibited activity against the chloroquine-sensitive *Plasmodium falciparum* 3D7 strain, while compounds **1**, **3**–**7**, and **11** were active against the chloroquine-resistant FCR-3 strain. This higher activity against the resistant strain is particularly encouraging, as it addresses a critical challenge in malaria treatment—resistance to commonly used antimalarial drugs like chloroquine (CQ) [60], which inhibits heme polymerase in malarial trophozoites, preventing heme conversion to hemazoin, leading to toxic heme accumulation and parasite death [61]. Compound **12** showed lower activity against the FCR-3 strain (IC_50_ > 1 µM) compared to that of 3D7 strain (IC_50_ = 0.80 µM). This difference could be attributed to the genetic variability between these strains. Additionally, we recognized that other compounds in the series outperformed compound **12** against the FCR-3 strain, suggesting that compound **12** may have limitations in its antiplasmodial activity spectrum.

Compound **3** emerged as the most potent antiplasmodial agent, displaying an impressive IC_50_ value of 0.0049 μg/mL against 3D7 strain, significantly outperforming by more than 10-fold the effect of chloroquine (IC_50_ = 0.06 μg/mL). Compounds **1** and **12** also showed promising activity against this strain with IC_50_ values of 0.16 μg/mL and 0.80 μg/mL, respectively. Against the FCR-3 strain, compound **3** also stands out, with an IC_50_ value of 0.12 μg/mL, while the positive standard CQ has an IC_50_ of 0.33 μg/mL. Such remarkable activity by compound **3** is highlighted when contrasted with results reported by Patel et al. [59]. Indeed, these authors showed that 2-amino and 2-phenylaminonaphthoquinones, display potent antiplasmodial activity against the 3D7 strain with IC_50_ values of 5.14 and 6.30 μM, respectively (Figure 2). By comparing such effects at equimolar concentrations, the IC_50_ value of compound **3** is 0.0186 μM against 3D7 strain, which is an enhancement of antiplasmodial activity by more than 250-fold.

Hydrophobicity (LogP), molar refractivity (CMR), and half-wave potential (E^I^_1/2_,) are three standard molecular descriptors commonly used in the structure–activity relationships of cytotoxic 1,4-naphthoquinones [62,63,64,65], which are key for delineating a large number of receptor-ligand interactions crucial to biological processes [66]. The half-wave potential data in Table 5 indicate that the insertion of phenylamino substituents into the naphthoquinone ring induces a shift in the half-wave potential toward more negative values compared to naphthoquinone ring-containing –Cl substituents. This suggests that this descriptor can partially explain the variations in antibacterial and antiplasmodial activities among the different phenylaminonaphthoquinone derivatives, providing a framework for further optimization of these compounds. Since compound **3** has a more positive half-wave potential value than the other compounds and exhibits the best antibacterial and antiplasmodial activities, it is tempting to suggest that it causes membrane disruption via an induction of oxidative stress.

The docking experiments showed that compounds **1**–**12** exhibited stronger binding interactions with clumping factor A, a virulence factor of *Staphylococcus aureus* that binds to fibrinogen [67,68]. These interactions were particularly pronounced among the halogenated derivatives (2-chloro-3-phenylaminonaphthoquinones), suggesting a potential mechanism of action against this pathogen. For antiplasmodial activity, the compounds exhibited best interactions with dihydroorotate dehydrogenase (DHODH) of *Plasmodium falciparum*, which catalyzes the rate-limiting step in the pyrimidine biosynthetic pathway and represents a potential target for antimalarial therapy [69]. These interactions were more pronounced than values obtained with atovaquone-inhibited cytochrome bc1 complex, and they consistently displayed higher binding affinities for both proteins compared to chloroquine. The docking calculations for DHODH revealed free binding energy values largely due to hydrogen bond acceptor interactions from the naphthoquinone ring and Van der Waals interactions from aromatic substituents. These results underscore the potential of phenylaminonaphthoquinone compounds, especially compound **3**, as promising candidates for antibacterial and antiplasmodial drug development, supporting the need for further investigation through detailed biological assays.

Pharmacokinetic and toxicity properties significantly influence the discovery and development of drugs. In this regard, computational models are viable alternatives to experimental methods. The ADMET analysis highlighted the favorable pharmacokinetic properties of the compounds studied (compounds **1**–**12**). All compounds, including chloroquine as a positive control, exhibited a molecular weight below 500 g/mol, which enhanced their drug-likeness and cellular penetrability [70]. High Caco-2 permeability and intestinal absorption rates suggest good oral bioavailability [71]. An adequate skin penetration was observed, although some compounds may require formulation optimization for effective transdermal delivery. Molecules will have difficulty penetrating the skin if the logKp value is greater than −2.5 cm/hour [72]. Most compounds show limited blood–brain barrier penetration, potentially reducing CNS side effects [73]. The analysis also revealed a low risk of drug-drug interactions, especially for compound **3**, which was highly active and did not inhibit CYP3A4. Furthermore, all compounds reflect moderate toxicity. These findings suggest that these phenylaminonaphthoquinones, especially compound **3**, could serve as lead compounds for the development of new antimalarial drugs, particularly against chloroquine-resistant strains.

## 4. Materials and Methods

### 4.1. Chemistry

#### 4.1.1. General

All the solvents and reagents were purchased from different companies, such as Aldrich (St. Louis, MO, USA) and Merck (Darmstadt, Germany), and were used as supplied. Melting points were determined on a Stuart Scientific SMP3 (Staffordshire, UK) apparatus and are uncorrected. Silica gel Merck 60 (230–400 mesh, from Merck) was used for supported, preparative column chromatography, and thin layer chromatography (TLC) aluminum foil 60F_254_ was used for analytical thin layer chromatography.

#### 4.1.2. Synthesis of Phenylaminonaphthoquinones with Solvent-Free

A mixture of 1,4-naphthoquinones or 2,3-dichloro-1,4-napthoquinone (1 mmol) and the respective arylamines (1 mmol) was supported on silica gel (0.2 g) and thoroughly mixed in an agate mortar, followed by grinding until the reaction was complete, as indicated by TLC. The solid mixture was filtered through a short column and eluted with petroleum ether/EtOAc. The spectral data of these compounds agree with the data reported in the literature [37].

#### 4.1.3. Experimental Log P

The determination of log P by the shake-flask method [74] was performed using a mixture of n-octanol and water in equal volumes, which were shaken at a temperature of 25 ± 1 °C for 24 h on a mechanical shaker to promote phase separation. The mixture was then allowed to stand for 24 h at the same temperature for further phase separation. A stock solution with a concentration of 0.01 mol per liter in water, pre-saturated with n-octanol, was prepared for each compound. Duplicate test vessels containing a mixture of the stock solution and n-octanol (pre-saturated with water) were shaken on a vortex shaker for 5 min. Phase separation was achieved by centrifugation of the vessels for 3 min at 4000× *g* (25 ± 1 °C), and the concentration of the solute in the water phase (C water) was measured from their absorbance at 205 nm using a UV spectrophotometer. The partition coefficient was calculated using the following equation.
Log Po/w = Log(C n-octanol/C water)

All the measurements were carried out at room temperature (25 ± 1 °C). In order to check the performance of the method, aniline was used as a reference substance.

#### 4.1.4. Molecular Descriptors

Calculation of lipophilicity (ClogP) and molar refractivity (CMR) was assessed using the ChemBioDraw Ultra 11.0 software, and the obtained values are shown in Table 5. Redox potentials of phenylaminonaphtoquinones **1**–**12** were measured using cyclic voltammetry at room temperature in acetonitrile as solvent using a platinum electrode and 0.1M tetraethylammonium tetrafluoroborate as the supporting electrolyte [37].

### 4.2. Biological Evaluation

#### 4.2.1. In Vitro Antimicrobial Screening

##### Bacterial Strains

The microorganisms used were *Staphylococcus aureus* (ATCC 25923), *Proteus mirabilis* (ATCC 12453) *Enterococcus faecalis* (ATCC 29212), *Escherichia coli* (ATCC 25922), *Salmonella typhimurium* (ATCC 19585), and *Klebsiella pneumoniae* (ATCC 700603).

##### Determination of the Minimum Inhibitory Concentration (MIC)

The antimicrobial activity of the phenylaminonaphthoquinones **1**–**12** was determined by the broth dilution method [75]. The following concentrations were tested: 200, 100, 50, 25, and 12.5 μg/mL. After incubation, the microbial growth was examined. The results are expressed in minimum inhibitory concentration (MIC), the lowest concentration of phenylaminonaphthoquinones with no visible growth. The bactericidal/bacteriostatic activity was determined by sub-cultivation of the samples in normal culture media at appropriate temperatures and incubation times. The MIC of each compound was performed in triplicate. The compounds **1**–**12** were dissolved in DMSO. Cefazolin and cefotaxime were used as positive controls at the same compound concentrations.

#### 4.2.2. Antiplasmodial Screening

The antimalarial activity assay was performed using *Plasmodium falciparum* 3D7 (chloroquine-sensitive) and FCR-3 (chloroquine-resistant) reference strains, which were cultured in vitro following the method described by Trager et al. [76] with some modifications.

The experiments were performed in 96-well culture plates, where the stock solutions of compounds **1**–**12** (1, 0.1, 0.01, and 0.001 µg/mL) were tested. A total of 100 µL of each product was distributed in triplicate on the culture plates, and 100 µL of a red blood cell suspension (4% hematocrit, final parasitemia of 1% ring stage) was added, resulting in a final volume of 200 µL. The test plates were incubated at 37 °C for 48 h [77], and parasitemia was determined by flow cytometry (BDFACScalibur). Erythrocytes were taken from each well culture plate, and 100 µL of ethidium bromide (10 µg/mL) was added and incubated for 30 min. A total of 100,000 cells per well culture plate were acquired and analyzed [78,79,80]. The IC_50_ of the pure products was determined by a regression curve of the percentage inhibition of parasite activity as a function of the logarithmic dose. Chloroquine diphosphate was used as the reference drug.

### 4.3. Computational Methods

The ligand–protein affinity has been studied using the molecular docking method [81] with proteins involved in antibacterial and antiparasitic were performed on compounds **1**–**12**. In order to understand the experimentally observed activity of the ligands we have selected the following proteins: Clumping Factor A from *Staphylococcus aureus* [82] (PDB ID: 1N67), atovaquone-inhibited cytochrome bc1 complex [71] (PDB ID: 4PD4), and *Plasmodium falciparum* dihydroorotate dehydrogenase [83] (PDB ID: 5FI8). The data for this analysis were obtained from the Protein Data Bank [84]. The molecular mechanics optimization was performed using the universal force field [85] (UFF) implemented in the Avogadro 1.97.0 program [86], and the corresponding SMILES [87] can be found in the Appendix A (Appendix A and Appendix A). The proteins were prepared using the UCSF CHIMERA 1.17.3 software [88], and the docking calculations were carried out through the CB-DOCK2 server using a fully automated blind docking approach [89,90]. The binding sites were identified using the CurPocket method [91,92], which employs a protein-surface curvature-based cavity detection approach. The free binding energies were calculated using the AutodockVina 1.0.2 program [93]. Results were compared with available literature data [83,94,95,96] to ensure the binding sites were appropriate for our study. The visualization was done using the Biovia Discovery Studio visualizer [97].

### 4.4. ADMET Prediction

pkCSM online tool (http://biosig.lab.uq.edu.au/pkcsm/prediction, accessed on 26 March 2024) [98] was used to predict absorption, distribution, metabolism, excretion, and toxicity (ADMET) of compounds **1**–**12**.

### 4.5. Statistical Analysis

All of the experiments were performed in two independent assays. The GraphPad Prism 8.0.2 software (San Diego, CA, USA) was utilized for the analysis. The results are expressed as the mean ± standard deviation (SD).

## 5. Conclusions

A solvent-free and silica gel-supported method for C–N bond formation via Michael-type addition offers a green and efficient alternative to traditional synthesis methods, improving both reaction time and yield. The synthesized phenylaminonaphthoquinones exhibited promising antibacterial activity against *Staphylococcus aureus*, especially compounds **1**, **3**, and **5**, while showing no activity against Gram-negative bacteria. Additionally, these compounds demonstrated notable antiplasmodial activity, particularly against chloroquine-resistant *Plasmodium falciparum* strains, with compound **3** emerging as a potent candidate for the development of novel antibacterial and antimalarial drugs.

## Figures and Tables

**Figure 1 ijms-25-10670-f001:**
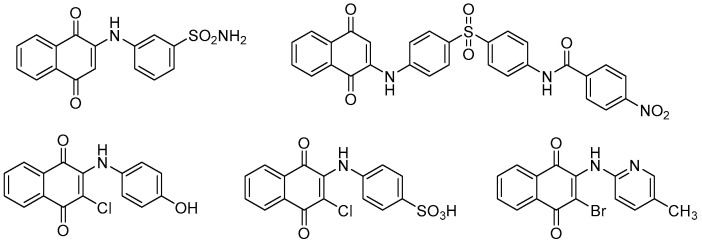
Important nitrogen-containing naphthoquinones hold interesting antimicrobial properties.

**Figure 2 ijms-25-10670-f002:**
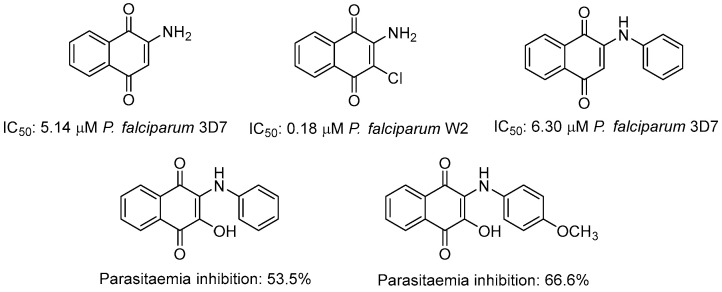
Structures of some important amino/phenylaminonaphthoquinones derivatives with antimalarial activity (IC_50_ and parasitemia inhibition) in literature.

**Figure 3 ijms-25-10670-f003:**
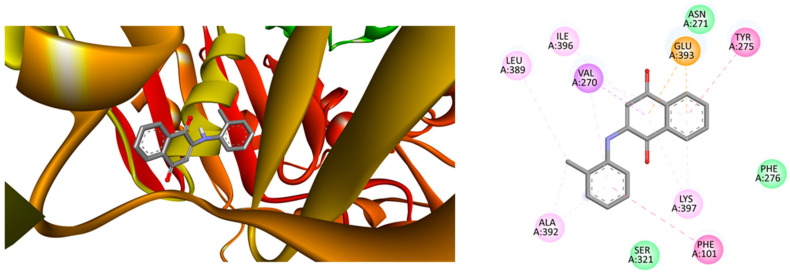
(**Left**) Best position obtained for compound **3**—clumping factor A from *Staphylococcus aureus* complex and (**Right**) 2D representation of the interactions between ligand and residues. Hydrogen atoms have been omitted in some cases for clarity. Green = conventional hydrogen bond; light green: Van der Waals interactions; purple = π (Aromatic) − σ interactions; orange = π − anion; and pink = alkyl − or π − alkyl interactions.

**Figure 4 ijms-25-10670-f004:**
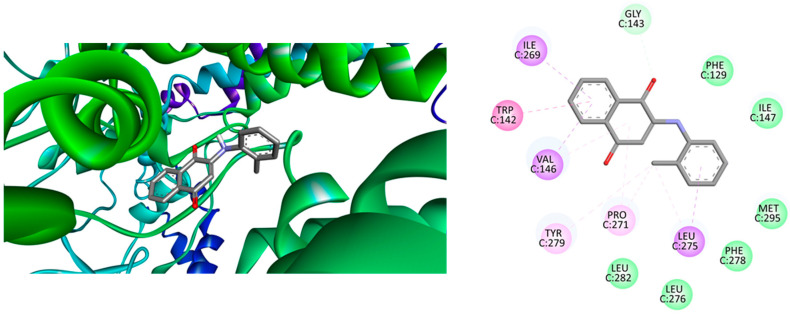
(**Left**) Best position obtained for Compound **3**—atovaquone-inhibited cytochrome bc1 complex and (**Right**) 2D representation of the interactions between ligand and residues. Hydrogen atoms have been omitted in some cases for clarity. Green = conventional hydrogen bond; light green: Van der Waals interactions; purple = π (Aromatic) − σ interactions; and pink = alkyl − or π − alkyl interactions.

**Figure 5 ijms-25-10670-f005:**
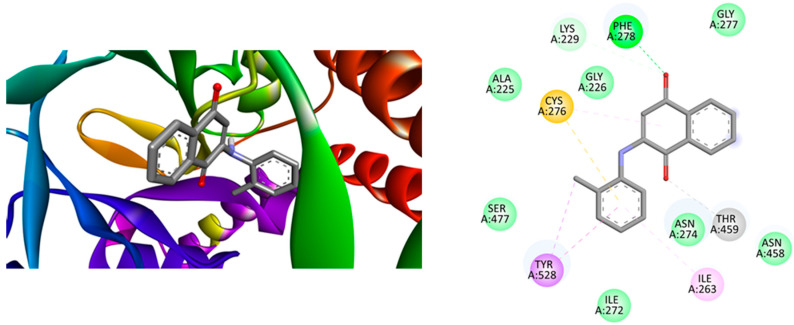
(**Left**) Best position obtained for Compound **3**—dihydroorotate dehydrogenase complex and (**Right**) 2D representation of the interactions between ligand and residues. Hydrogen atoms have been omitted in some cases for clarity. Green = conventional hydrogen bond; light green: Van der Waals interactions; purple = π (Aromatic) − σ interactions; and pink = alkyl − or π − alkyl interactions.

**Table 1 ijms-25-10670-t001:** Comparison of yields and time reactions of phenylaminonaphthoquinones **1–12** synthesized with solvent and solvent-free.

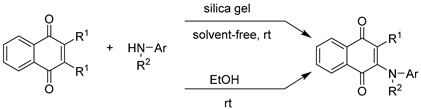							
Product N°	R^1^	R^2^	Aryl	Solvent-Free		Solvent (EtOH)	
				Yield (%)	Time	Yield (%)	Time
**1**	H	H	Ph	90	2.0 h	95	5.0 h
**2**	Cl	H	Ph	96	2.0 h	97	5.0 h
**3**	H	H	2-MePh	85	2.5 h	87	6.0 h
**4**	Cl	H	2-MePh	93	2.5 h	93	6.0 h
**5**	H	H	4-HOPh	89	0.5 h	92	2.0 h
**6**	Cl	H	4-HOPh	90	0.5 h	94	2.0 h
**7**	H	H	4-MeOPh	87	1.5 h	84	3.0 h
**8**	Cl	H	4-MeOPh	92	1.5 h	89	3.0 h
**9**	H	H	2,5-(MeO)_2_Ph	91	2.0 h	85	5.0 h
**10**	Cl	H	2,5-(MeO)_2_Ph	86	2.0 h	93	5.0 h
**11**	H	Me	Ph	72	5.0 h	78	12.0 h
**12**	Cl	Me	Ph	86	5.0 h	89	12.0 h

Structural characterization of compounds **1**–**12** have been reported [37].

**Table 2 ijms-25-10670-t002:** PASS prediction results for compound **3**.

Pa	Pi	Activity
0.800	0.012	Antineoplastic
**0.700**	**0.039**	**Membrane permeability inhibitor**
0.636	0.026	Oxidoreductase inhibitor
**0.536**	**0.007**	**Para amino benzoic acid antagonist**
0.466	0.025	1-Acylglycerol-3-phosphate O-acyltransferase inhibitor
0.316	0.083	Antiprotozoal (Trypanosoma)
**0.312**	**0.093**	**Anti-infective**
0.259	0.092	Cytochrome-b5 reductase inhibitor
0.215	0.114	Glycerol-3-phosphate dehydrogenase inhibitor
**0.192**	**0.010**	**Antibiotic Anthracycline-like**
**0.174**	**0.084**	**Antiprotozoal (Plasmodium)**
**0.162**	**0.153**	**Antibacterial**

**Table 3 ijms-25-10670-t003:** Minimum inhibitory concentration (MIC) of phenylaminonaphthoquinones **1**–**12**.

MIC (μg/mL)						
Compound	*S. aureus* G+	*P. mirabilis* G+	*E. faecalis* G+	*E. coli* G−	*S. typhimurium* G−	*K. pneumonia* G−
**1**	3.9	40.0	49.9	26.3	38.8	>50
**2**	36.5	39.9	>50	>50	>50	>50
**3**	3.8	19.9	50.0	25.5	13.2	26.3
**4**	>50	40.9	37.1	>50	32.5	44.6
**5**	3.2	24.8	>50	>50	>50	>50
**6**	19.6	29.9	49.9	39.8	20.0	40.0
**7**	5.7	18.4	31.2	>50	29.9	37.3
**8**	21.9	52.6	40.1	>50	49.9	49.1
**9**	>50	29.9	38.2	>50	43.8	>50
**10**	4.7	>50	70.8	23.5	31.7	>50
**11**	5.5	44.0	30.2	>50	>50	>50
**12**	29.0	11.5	47.2	28.6	50.0	>50
CFZ	4.2	5.6	8.4	>50	7.1	8.5
CTX	8.9	8.9	10.9	>50	8.2	8.4

G+: Gram-positive; G−: Gram-negative. Positive control: CFZ, cefazolin; CTX, cefotaxime.

**Table 4 ijms-25-10670-t004:** Antimalarial activity of compounds **1**–**12** against *Plasmodium falciparum*.

Compound	*Plasmodium falciparum* 3D7		*Plasmodium falciparum* FCR-3	
	Growth Inhibition (%)	IC_50_ (μg/mL)	Growth Inhibition (%)	IC_50_ (μg/mL)
**1**	58.0 ± 3.0	0.16	65.0 ± 1.0	0.74
**2**	37.0 ± 1.0	>1	13.0 ± 2.0	> 1
**3**	98.0 ± 1.0	0.0049	97.0 ± 0.2	0.12
**4**	42.0 ± 2.0	>1	56.0 ± 1.0	0.86
**5**	18.0 ± 1.0	>1	61.0 ± 2.0	0.78
**6**	41.0 ± 2.0	>1	56.0 ± 3.0	0.87
**7**	49.0 ± 2.0	>1	57.0 ± 1.0	0.88
**8**	26.0 ± 2.0	>1	34.0 ± 2.0	>1
**9**	33.0 ± 1.0	>1	24.0 ± 2.0	>1
**10**	20.0 ± 3.0	>1	21.0 ± 4.0	>1
**11**	26.0 ± 8.0	>1	72.0 ± 2.0	0.55
**12**	62.0 ± 5.0	0.80	25.0 ± 5.0	>1
**CQ**	-	0.06	-	0.33

CQ: chloroquine.

**Table 5 ijms-25-10670-t005:** Molecular descriptors: theoretical and experimental Log P, molar refractivity, and half-wave potential (E^I^_½_) values of phenylaminonaphthoquinones **1**–**12**.

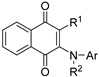
Product N°	R^1^	R^2^	Aryl	Log PTheor ^a^	Log PExp	CMR (cm^3^/mol) ^a^	−E^I^_1/2_ (mV) ^b^
**1**	H	H	Ph	1.55	1.42 ± 0.21	7.1403	770
**2**	Cl	H	Ph	1.59	1.61 ± 0.11	7.6317	618
**3**	H	H	2-MePh	2.04	1.89 ± 1.12	7.6041	727
**4**	Cl	H	2-MePh	2.08	2.20 ± 0.13	8.0955	640
**5**	H	H	4-HOPh	1.16	1.17 ± 0.81	7.2934	775
**6**	Cl	H	4-HOPh	1.20	1.22 ± 0.13	7.7848	690
**7**	H	H	4-MeOPh	1.43	1.39 ± 0.15	7.7572	856
**8**	Cl	H	4-MeOPh	1.46	1.81 ± 1.04	8.2486	658
**9**	H	H	2,5-(MeO)_2_Ph	1.30	1.60 ± 0.23	8.3741	767
**10**	Cl	H	2,5-(MeO)_2_Ph	1.34	1.94 ± 0.56	8.8655	614
**11**	H	Me	Ph	2.34	2.31 ± 0.30	7.6041	760
**12**	Cl	Me	Ph	2.38	2.33 ± 0.02	8.0955	530

^a^ Log P and CMR were calculated using the ChemBioDraw Ultra 11.0 software. ^b^ −E^I^_½_ (mV) was measured using cyclic voltammetry [37].

**Table 6 ijms-25-10670-t006:** Free binding energy results (in kcal·mol^−1^) from molecular docking calculations of phenylaminonaphthoquinones **1**–**12** with antimicrobial protein clumping factor A from *Staphylococcus aureus* (1N67); and antiparasitic proteins such as atovaquone-inhibited cytochrome bc1 complex (4PD4) and dihydroorotate dehydrogenase (5FI8).

Compound	ΔE_Bind_ (kcal·mol^−1^)		
	Antibacterial	Antiparasitic	
	1N67	4PD4	5FI8
**1**	−9.0	−9.2	−10.2
**2**	−9.3	−8.9	−9.3
**3**	−9.4	−9.2	−9.1
**4**	−10.4	−8.7	−9.2
**5**	−9.3	−8.8	−10.0
**6**	−10.0	−8.4	−9.9
**7**	−9.1	−9.0	−9.4
**8**	−9.3	−8.5	−10.1
**9**	−9.1	−8.6	−9.2
**10**	−9.0	−8.5	−9.3
**11**	−9.1	−9.3	−9.6
**12**	−9.3	−8.7	−9.3
Chloroquine	----	−6.9	−7.3

**Table 7 ijms-25-10670-t007:** ADMET properties of chemical of phenylaminonaphthoquinones **1**–**12**.

	Properties									
	A			D			M	E	T	
	Model Name									
N°	Caco-2	IA	SP	VDss	BBB	CNS	CYP2D6/ CYP3A4 Inhibitor	TC	Oral Rat Acute Tox. (LD_50_)	Oral Rat Chronic Tox. -LOAEL
**1**	1.341	93.387	−2.778	0.131	0.239	−0.844	No/Yes	0.169	2.478	2.051
**2**	1.384	92.986	−2.762	0.179	0.200	−0.814	No/Yes	0.054	2.460	1.903
**3**	1.341	93.876	−2.780	0.216	0.239	−0.844	No/No	0.181	2.470	2.019
**4**	1.384	93.475	−2.763	0.257	0.216	−0.815	No/Yes	0.011	2.457	1.859
**5**	0.893	92.399	−2.769	0.048	−0.068	−2.013	No/No	0.139	2.270	1.962
**6**	0.979	91.999	−2.754	0.059	−0.106	−1.929	No/No	−0.065	2.629	1.373
**7**	1.311	94.436	−2.768	0.147	0.045	−2.014	No/Yes	0.186	2.572	1.932
**8**	1.354	94.035	−2.757	0.174	0.007	−1.930	No/Yes	0.082	2.644	1.496
**9**	0.989	96.100	−2.991	0.192	−0.184	−2.211	No/Yes	0.238	2.738	1.702
**10**	1.078	95.687	−2.819	0.196	−0.207	−2.123	No/Yes	0.434	2.937	1.514
**11**	1.816	97.425	−2.141	0.288	0.269	−1.404	No/Yes	0.261	1.918	1.815
**12**	1.834	97.025	−2.502	0.434	0.212	−1.375	No/Yes	0.131	2.198	1.690
CQ	1.257	90.072	−2.765	1.464	0.482	−2.369	Yes/Yes	1.118	2.750	0.778

Caco-2: Caucasian colon adenocarcinoma permeability (Log Papp in 10^−6^ cm/s). IA: intestinal absorption (% Absorbed). SP: skin permeability (logKp). VDss: steady-state volume of distribution (Log L/kg). BBB: blood–brain barrier permeability (log BB). CNS: central nervous system (Log PS). CYP2D6: Cytochrome P_450_ 2D6 inhibitor; CYP3A4: Cytochrome P_450_ 3A4 inhibitor. TC: total clearance (Log mL/min/kg). LD_50_: lethal dose, 50% (mol/Kg). LOAEL: lowest observed adverse effect level (Log mg/kg bw/day). CQ: chloroquine.

## Data Availability

Data are contained within the article and Appendix A.

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
