# Peer review of "Exploring the Antibacterial and Antiparasitic Activity of Phenylaminonaphthoquinones—Green Synthesis, Biological Evaluation and Computational Study"

_ijms, 2024, doi:10.3390/ijms251910670_

Round 1

Reviewer 1 Report

Comments and Suggestions for Authors

The authors have submitted an interesting article, which mainly reports a solvent-free synthetic method of phenylaminonaphthoquinones, and evaluates the antibacterial and antiparasitic activity, as well as computational study. The synthetic method is green with less reaction time compared with traditional methods. This work seems very meaningful, but some key data should be provided before it can be considered for publication:

1. Compound 1-12 were synthesized in the study. Please provide the characterization of them for proving their successful preparation.

2. The study reports a green synthetic method of phenylaminonaphthoquinones, I found the reaction was a grinding operation until Methods section (Section 4). There are two questions, during the reaction, is the grinding manual? How about the reaction temperature?

If grinding is at room temperature, or do not produce heat, the room temperature synthesis also belongs to a characteristic of green synthesis.

Moreover, it’s better to mention the grinding operation in the parts except Section 4.

3. Table 1, what about the reaction temperature using EtOH as solvent?

4. Figure 2, it is better to provide IC50 value if the literature has.

5. Section 4.2.1.2, “The following concentrations were tested: 200, 100, 50, 25, and 12.5 μg/mL.” Maybe smaller concentration should be added, i.e. 6.25, 3.1 μg/mL.   

6. Some compounds exhibited good antibacterial or antiparasitic activity, e.g. 3 and 11. How about the toxicity towards human cells?

Author Response

Reviewer 1: The authors have submitted an interesting article, which mainly reports a solvent-free synthetic method of phenylaminonaphthoquinones, and evaluates the antibacterial and antiparasitic activity, as well as computational study. The synthetic method is green with less reaction time compared with traditional methods. This work seems very meaningful, but some key data should be provided before it can be considered for publication:

We thanks the referee for his/her nice comment.

  1. Compound 1-12were synthesized in the study. Please provide the characterization of them for proving their successful preparation.

Answer: We appreciate the reviewer’s comment. In response, compounds 1-12 were previously synthesized by our research group and their synthesis are described in detail in our article published in the European Journal of Medicinal Chemistry (2010, 45, 6052-6057; quoted as reference 37 in the manuscript). We have added a paragraph in the Results section to underline that compounds 1-12 have been already synthesized by our group in previous work.

  1. The study reports a green synthetic method of Phenylaminonaphthoquinones; I found the reaction was a grinding operation until Methods section (Section 4). There are two questions, during the reaction, is the grinding manual? How about the reaction temperature?

If grinding is at room temperature, or do not produce heat, the room temperature synthesis also belongs to a characteristic of green synthesis.

Moreover, it’s better to mention the grinding operation in the parts except Section 4.

Answer: We thank the reviewer for the comments. The grinding operation is conducted at room temperature and do not produce heat. We have included in the section 2.1 a paragraph highlighting that the green synthesis was performed by grinding manual.

  1. Table 1, what about the reaction temperature using EtOH as solvent?

Answer: We thank the reviewer's comments. The reaction was carried out at room temperature and the corresponding details have been added in Table 1.

  1. Figure 2, it is better to provide IC50 value if the literature has.

Answer: We appreciate the reviewer's comments. As far as we know, the literature does not provide the IC50 values of two compounds depicted in Figure 2.

  1. Section 4.2.1.2, “The following concentrations were tested: 200, 100, 50, 25, and 12.5 μg/mL.” Maybe smaller concentration should be added, i.e. 6.25, 3.1 μg/mL.  

 Answer: We thank the reviewer for the valuable suggestion. We agree that testing lower concentrations such as 6.25 and 3.1 μg/mL could provide further insight into the compound's activity at sub-inhibitory levels, but we prioritized the concentrations that were more likely to show measurable biological effects based on initial screening results. Owing to time constraints and resource limitations, we focused on a range that covered both the high and low ends of the expected activity spectrum. However, future work could explore these lower concentrations for a more comprehensive understanding of the dose-response relationship.

  1. Some compounds exhibited good antibacterial or antiparasitic activity, e.g. 3and 11. How about the toxicity towards human cells?

Answer: We agree that toxicity experiments in human cells are critical to evaluate the potential use of these compounds as treatment. Currently, we have not performed toxicity evaluations of compounds 3 and 11 in human cells. We have included a paragraph highlighting the relevance of toxicity experiments in human cells for future studies. However, we would like to emphasize that we evaluated the antitumor activity of these compounds using a panel of cancer cell lines (MCF7, DU145 and T24 cells) and non-transformed BALB/3T3 fibroblasts. In this assay, we obtained IC50 values of 6.2 and 43.3 µg/mL in BALB/3T3 cells for compounds 3 and 11, respectively. By calculating the C/T ratio (IC50 values in fibroblasts/IC50 values in tumor cells), we obtained values of 3.14 and 4.27 for compounds 3 and 11, respectively. Although these values do not directly measure toxicity, they indicate a degree of selectivity, suggesting that these compounds are less toxic to non-tumor cells compared to tumor (reference 37 in the manuscript).

Reviewer 2 Report

Comments and Suggestions for Authors

The manuscript entitled “Exploring the antibacterial and antiparasitic activity of phenylami-2 nonaphthoquinones. Green synthesis, biological evaluation and 3 computational study’has been reviewed.

The authors explored the antibacterial and antiparasitic activity of phenylami-2 nonaphthoquinones. Green synthesis, biological evaluation and 3 computational study.

However, the following observations were noted:

1.       A very crucial part of any manuscript on synthesis of compounds is missing. I did not see the spectra results/ structural elucidation ( eg IR, NMR, MS etc) of the synthesized compounds. This manuscript can only become publishable on the inclusion of the spectra analyses. The authors should include them before this manuscript can become meaningful. The spectra should be mentioned in the abstract section.

2.       Line 167-169, under table 4. The compounds should be presented in the decreasing order of antiplasmodial activities. It should be compounds 3, 12 and 1. Not the other way

3.       Line 170-171, I suggest that the authors should account for low performance of compound 12 (IC50 = ˃1) plasmodium faciparium FCR-3 compared to (IC50= 0.80) with plasmodium faciparium 3D7. Note most compounds outperformed compound 12 against FCR-3.

4.       I suggest that the authors add a paragraph on the structure-activity relationship (SAR) of some of the synthesized compounds. For example, there is a structural similarity amongst the best antiplasmodial compounds 1,3,11 and 12. Furthermore, structurally similar compounds 3 and 7 in exception to E faeculis and E coli respectively maintained significant antibacterial activities

5.       In table 6, if possible, the authors should account for some result discrepancies in the in vitro and molecular docking reports. For example, compounds 1,3 and 12 were best antiplasmodial agents in vitro, however, in molecular docking study, compound 11 (4PD4 - -9.3 kcal/mol). Compounds 1, 8 and 5 (5F18) were the best

Conclusion: This work is a synthesis work not a virtual screening work, as such needs structural elucidation. Any work on synthesis of new compounds must go with FTIR, NMR, MS or elemental analysis to prove that the said compounds were actually synthesized. Unfortunately, this is not found in the manuscript.

Author Response

Reviewer 2:

The manuscript entitled “Exploring the antibacterial and antiparasitic activity of phenylami-2 nonaphthoquinones. Green synthesis, biological evaluation and 3 computational study’’ has been reviewed.

The authors explored the antibacterial and antiparasitic activity of phenylami-2 nonaphthoquinones. Green synthesis, biological evaluation and 3 computational study.

However, the following observations were noted:

  1. A very crucial part of any manuscript on synthesis of compounds is missing. I did not see the spectra results/ structural elucidation (e.g IR, NMR, MS etc) of the synthesized compounds. This manuscript can only become publishable on the inclusion of the spectra analyses. The authors should include them before this manuscript can become meaningful. The spectra should be mentioned in the abstract section.

Answer: We appreciate the reviewer’s comment. In response, compounds 1-12 were previously synthesized by our research group and their synthesis are described in detail in our article published in the European Journal of Medicinal Chemistry (2010, 45, 6052-6057; quoted as reference 37 in the manuscript). We have added a paragraph in the Results section to underline that compounds 1-12 have been already synthesized by our group in previous work.

  1. Line 167-169, under table 4. The compounds should be presented in the decreasing order of antiplasmodial activities. It should be compounds 3, 12 and 1. Not the other way

Answer: We thank the reviewer for the valuable suggestion. We have included under table 4 section a paragraph including her/his suggestion.

  1. Line 170-171, I suggest that the authors should account for low performance of compound 12 (IC50 = Ëƒ1) plasmodium faciparium FCR-3 compared to (IC50= 0.80) with plasmodium faciparium 3D7. Note most compounds outperformed compound 12 against FCR-3.

Answer: We appreciate the reviewer’s observation of the performance of compound 12. As noted, compound 12 showed lower activity against the FCR-3 strain (IC50 > 1 µM) compared to the 3D7 strain (IC50 = 0.80 µM). We have included in the discussion section a paragraph including the suggestion.

  1. I suggest that the authors add a paragraph on the structure-activity relationship (SAR) of some of the synthesized compounds. For example, there is a structural similarity amongst the best antiplasmodial compounds 1,3,11 and 12. Furthermore, structurally similar compounds 3 and 7 in exception to E faeculis and E coli respectively maintained significant antibacterial activities

Answer: We thank the suggestion of the reviewer. We have included in Discussion section a paragraph including her/his suggestion.

  1. In table 6, if possible, the authors should account for some result discrepancies in the in vitro and molecular docking reports. For example, compounds 1,3 and 12 were best antiplasmodial agents in vitro, however, in molecular docking study, compound 11 (4PD4 - -9.3 kcal/mol). Compounds 1, 8 and 5 (5F18) were the best.

Answer: We thank the reviewer for the insightful comments. Although compounds other than compound 3 showed comparable or slightly better molecular docking results, compound 3 was selected based on its superior experimental antiplasmodial activity. Discrepancies between experimental activity and docking interaction energies often arise because docking simulations cannot account for all the experimental variables that influence a given biological activity. Moreover, although docking studies offer valuable qualitative insights into potential intermolecular interactions, they do not always directly correlate with biological activity. Therefore, compounds with the best docking scores may not necessarily exhibit the highest experimental activity but still provide important information regarding the interactions under study.

Conclusion: This work is a synthesis work not a virtual screening work, as such needs structural elucidation. Any work on synthesis of new compounds must go with FTIR, NMR, MS or elemental analysis to prove that the said compounds were actually synthesized. Unfortunately, this is not found in the manuscript.

Answer: We are grateful to the referee for the suggestions. In the revised version, we have incorporated her/his recommendations. We hope that changes we made have enhanced the quality of the manuscript.

Round 2

Reviewer 1 Report

Comments and Suggestions for Authors

Agree

Reviewer 2 Report

Comments and Suggestions for Authors

Significant corrections have been made in the revised manuscript. The quality of the manuscript has been improved.